# Novel Control System Strategy for the Catalytic Oxidation of VOCs with Heat Recovery

**Angel Federico Miranda** [1,*]📵, **María Laura Rodríguez** [2]📵, **Federico Martin Serra** [2] **and Daniel Oscar Borio** [1]

[1] Planta Piloto de Ingeniería Química (PLAPIQUI-UNS/CONICET), Camino La Carrindanga km. 7, Bahía Blanca 8000, Argentina; dborio@plapiqui.edu.ar
[2] Instituto de Investigación en Tecnología Química (INTEQUI-UNSL/CONICET), Almirante Brown 1455, San Luis 5700, Argentina; mlrodri@unsl.edu.ar (M.L.R.); fserra@ieee.org (F.M.S.)
[*] Correspondence: amiranda@plapiqui.edu.ar

**Abstract:** A theoretical study of the dynamic closed-loop behaviour of a reactor/feed-effluent heat exchanger (FEHE)/furnace system for the catalytic combustion of volatile organic compounds (VOCs) is presented. A 1D pseudohomogeneous plug-flow model is proposed to simulate the non-steady-state operation of the monolith reactor and the FEHE, while the furnace behaviour is described by means of a heterogeneous model of lumped parameters. Positive energy feedback is a source of instability that leads to strong thermal oscillations (limit cycles) and may cause damage to the equipment and sintering of the catalyst. The design of a robust and flexible control system and an efficient control strategy are, therefore, required to ensure safe and stable operation. The response of the system under three different control strategies to the most frequent disturbance variables—the feed flowrate ($F_{V0}$) and feed concentration of VOCs ($C_{0Et}$)—was evaluated. One of the control strategies consisted of a single-loop feedback system with servomechanism changes in the reactor inlet temperature ($T_0$) that manipulated the bypass valve and, sequentially, the natural gas flowrate in the furnace ($F_{NG}$). This approach made it possible to meet the control objective (reducing VOCs) without losing controllability and while minimizing the use of external fuel.

**Keywords:** VOC emissions; catalytic oxidation; heat-integrated system; control strategies; feedforward control; advanced control system

## 1. Introduction

Volatile organic compounds (VOCs) are gases that are emitted into the air from products or processes. Due to their properties, they are present in solvents, solvent-based paints and varnishes, glues, dispersants, degreasing agents, lubricants, and liquid fuels, and they are emitted from industries that synthesize them, generate them as by-products, or use them in their processes.

Due to their negative effects and widespread use, there is a clear need to avoid/reduce their emission into the environment. Worldwide, there are strict environmental regulations [1,2] that establish limits on the maximum concentrations of VOCs allowed to be vented into the air (the emission limit value (ELV)).

Catalytic oxidation is emerging as a promising technology for their reduction/elimination, particularly when: (a) the concentration of VOCs is relatively low and, therefore, recovery is economically unfeasible and (b) the concentration or flowrate of pollutants is not constant over time and, consequently, a versatile system capable of adapting to different temporal emission patterns is required [3].

These end-of-pipe systems have a twofold objective: to keep the concentration below the ELV and to use the recovered heat to reduce the amount of fuel necessary to preheat large volumes of contaminated air up to the reaction temperature. This, in turn, serves a dual purpose: (i) to reduce the energy demand and, thus, the associated operating costs

and (ii) to reduce the $CO_2$ emissions resulting from the use of fossil fuels in the removal process. Thus, the heat generated in the combustion of VOCs can be partially recovered through a feed-effluent heat exchanger (FEHE) to preheat the feed stream using the hot gases leaving the reactor. The heat needed to reach the reaction temperature can then be generated in a furnace where natural gas is admitted [4].

Energy recovery introduces positive feedback structures into the system, which may dramatically alter the time constants of the plant [5] and result in a variety of steady-state and dynamic phenomena, such as the snowball effect, extremely sluggish responses, oscillatory behaviour (limit cycles), and even instability [5–8].

Bildea et al. [9] studied a coupled reactor/separation/recycle system for toluene hydrodealkylation (HDA) and found that the interaction between reaction and separation through material recycling can lead to unfeasibility, steady-state multiplicity, and instability.

Morud and Skogestad [10] discussed the dynamics of an industrial multibed ammonia reactor where positive feedback due to heat integration led to oscillatory behaviour in the range from about 300 °C to 500 °C. The authors concluded that the physical cause for this somewhat unusual instability was a combination of the positive heat feedback in the preheater and the non-minimum phase behaviour (inverse response dynamics) of the reactor temperature response.

Luyben [11] described the dynamic problems that occur in reactor–FEHE systems and showed that the positive feedback of energy can produce an open-loop unstable process. However, the system can be made closed-loop stable through the use of an inlet temperature controller that bypasses cold material around the heat exchanger and mixes it with the heated stream to achieve the desired inlet reactor temperature.

Additional units are usually included in heat-integrated designs as follows: (i) Heater for the start-up. Since positive feedback due to heat integration may lead to state multiplicity, the heater duty can be manipulated in a temperature control loop to ensure stable operation; (ii) Steam generator. The energy introduced by the heater has to be removed; for example, by increasing the steam. Since the furnace is a heat source and the excess energy is rejected to a heat sink (steam generator), the reactor can be viewed as a heat pump [12].

Bildea and Dimian [13] studied the steady-state and dynamic behaviour of a heat-integrated PFR consisting of a feed-effluent heat exchanger (FEHE), a furnace, an adiabatic tubular reactor, and a steam generator. The system exhibited oscillatory behaviour with realistic values for the model parameters, and the selection of the FEHE efficiency was a critical step to achieve the desired steady state and ensure stability. The research showed a close relationship between design and controllability.

Several control structures have been proposed to control the reactor inlet temperature. Silverstein and Shinnar [14] studied the effects of design parameters on the dynamic stability of systems with an FEHE followed by a furnace before the adiabatic reactor. They recommended controlling the reactor inlet temperature with the furnace duty. They also explored bypassing cold material around the FEHE to provide an additional manipulated variable. For the HDA process, Terrill and Douglas [15] examined the use of multiple FEHEs in series to preheat the reactor feed with hot reactor effluents. In the process, a furnace is located before the reactor. Among the most common control structures, the authors suggested the control of the temperature of the mixed stream after the FEHE through the manipulation of the bypass flow together with the control of the reactor inlet temperature through the manipulation of the fuel admitted to the furnace [16]. If, as in the case of total closure of the bypass valve, the temperature rise in the FEHE is not sufficient, the fuel flow to the furnace could be manipulated for additional energy supply.

Luyben (2012) [16] studied different control structure configurations for the production of dimethyl ether (DME) from methanol, an exothermic and reversible vapour-phase reaction. The author explored configurations without a furnace and with a furnace with different percentages of bypass stream. Finally, he proposed a novel flowsheet and control structure. Instead of mixing the cold bypass with the hot stream from the FEHE, the bypass was mixed with the stream coming from the furnace, and this mixed stream was, in

turn, fed into the reactor. The stream passing through the FEHE was fed directly into the furnace. This setup permits tight control of the reactor inlet temperature, a key variable. In addition, a nonlinear feedforward (FF) control structure was employed to reject feed flowrate disturbances by manipulating the natural gas valve in order to reduce energy consumption in the furnace.

In the present study, the response of a reactor/FEHE/furnace system for the catalytic oxidation of $VOC_S$ under three different control system strategies to the two most frequent disturbances—changes in the process gas flowrate ($F_{V0}$) and the feed VOC concentration ($C_{0Et}$)—was evaluated.

Implementing an effective control strategy is crucial to (i) minimize out-of-specification periods (ELV requirement not fulfilled), (ii) avoid oversizing of the main equipment and reduce investment costs, (iii) minimize operating costs due to the external heat supply, and (iv) avoid severe thermal oscillations that can damage the catalyst and cause considerable stress on both the reactor and heat-exchanger materials.

## 2. Results and Discussion

### 2.1. Base Case

Figure 1 shows the axial temperature profiles along the monolithic reactor and the counter-current plate heat exchanger as a base case under steady-state conditions. The temperature downstream of the mixer ($T_{mix}$ see Figure 1) and the temperature rise caused by the furnace ($\Delta T_f$) are also indicated. The reactor inlet temperature ($T_0 = 185\ °C$) is reached by circulating 34% of the feed stream around the FEHE; that is, the selected value for $\lambda$ is 0.66. The preheated feed ($\lambda F_{V0}$) is then mixed with the bypass stream ($1 - \lambda)F_{V0}$ at $T_{in} = 20\ °C$. At the mixer, the temperature reaches about 144 °C. Inside the furnace, the mixed stream receives a constant external heat supply ($Q_f = 136$ kW), which increases the temperature of the process stream up to $T_0 = 185\ °C$. Figure 1 also shows the axial concentration profiles of ethanol and acetaldehyde (intermediary VOC compound). It can be observed that, for the base case, VOCs are completely consumed within around 70% (35 cm) of the total reactor length.

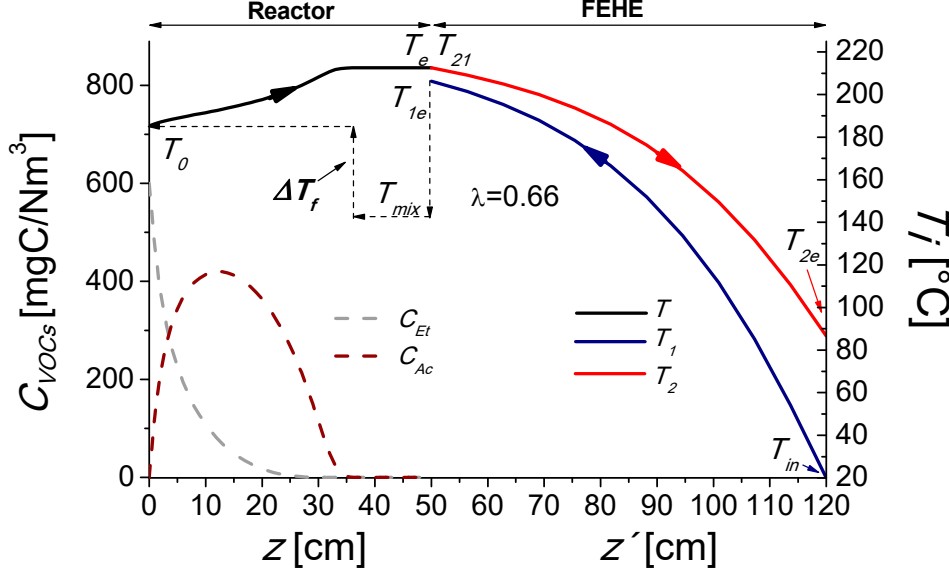

**Figure 1.** Axial temperature and concentration profiles within the reactor and temperature profiles within the FEHE. $T_0 = 185\ °C$, $C_{0Et} = 600$ mg C/$Nm^3$, $T_{in} = 20\ °C$, $F_{V0} = 10,000\ Nm^3/h$, $Q_f \approx 136$ kW ($\Delta T_f \approx 41\ °C$), $\lambda = 0.66$.

A typical increasing axial temperature profile can be observed in the reactor, showing a more pronounced slope in the zone where acetaldehyde is consumed, because the second reaction is more exothermic than the first one (compare $\Delta H_{r2}$ and $\Delta H_{r1}$ in Tables

of Section 3.2 Reactor, FEHE, and Furnace). When the VOCs are completely consumed and both reactions are fully extinguished, the reactor temperature reaches a plateau. This final temperature increase corresponds to the adiabatic temperature rise ($\Delta T_{ad}$), which is proportional to the VOC concentration in the feed stream. In the base case, the catalytic combustion of 600 mg C/Nm$^3$ of ethanol leads to a total temperature rise of ~27 °C.

In heat-integrated systems, recycling introduces positive feedback structures into the system, which may dramatically alter the time constants of the plant [5], resulting in a variety of steady-state and dynamic phenomena, such as oscillatory behaviour (limit cycles) and even instability [5,7,8]. The possible existence of multiple steady states (stable, metastable, and unstable) in this integrated system can be explained by the presence of two necessary conditions: nonlinearity and process feedback [4]. The inverse response is an additional phenomenon that impacts the system stability [10,17].

Figure 2 shows an S-shaped curve that relates the inlet reactor temperature ($T_0$) to the variable $\lambda$ under steady-state conditions. Steady-state multiplicity can be observed; that is, within a certain range of values for $\lambda$, the system can be operated at different of inlet reactor temperatures, as well as VOC conversion levels. Depending on the selected reactor inlet temperatures, the system can show qualitatively different open-loop dynamic behaviours; i.e., stable steady states at high and low conversion levels, sustained oscillations, or limit cycles and open-loop instability can be observed [18].

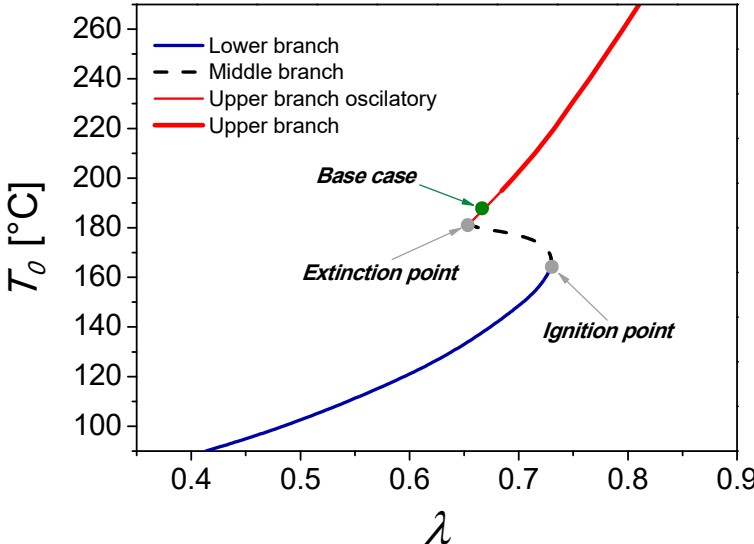

**Figure 2.** Inlet reactor temperature as a function of the fraction of the feed flowrate through the FEHE. Base-case operating conditions are those shown in Figure 1.

The operating condition for the chosen base case is located at the upper branch of the S-shaped curve at an inlet reactor temperature near the extinction point. The selection of this particular value for $T_0$ leads to total VOC abatement without excessive energy expenditure for the preheating of the process stream, avoiding excessively high reactor temperatures that could cause thermal stress in the catalyst and reactor materials.

Figure 3 shows the temporary evolution of the inlet ($T_0$) and outlet ($T_e$) reactor temperatures for the base case. No control action takes place; i.e., the system is operated under open-loop conditions. Although no disturbances occur, after 3000 s, a strong sustained oscillatory behaviour can be found in $T_0$ and $T_e$. Analogous behaviour can be observed for the temperatures inside the FEHE (results not shown in Figure 3). Notice that the reactor outlet temperature reaches a maximum of 270 °C cyclically every 200 s. This phenomenon has been analysed in previous work [18].

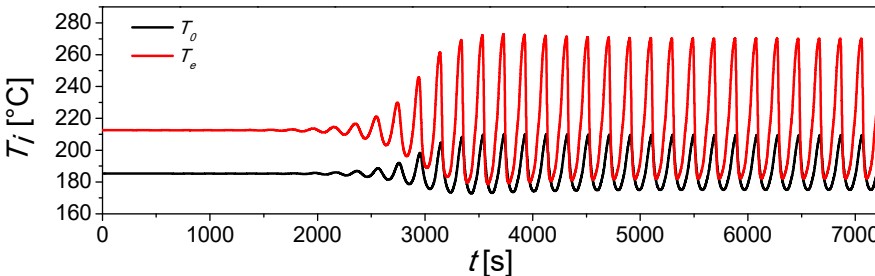

**Figure 3.** Open-loop temporal profiles for the reactor inlet temperature ($T_0$) and reactor outlet temperature ($T_e$) under the operating conditions for the base case shown in Figure 1.

In this situation, it is necessary to design a robust and flexible control system to ensure the abatement of the VOCs under safe and stable conditions, avoiding the detected strong limit cycles, which could certainly cause damage to the equipment and sintering of the catalyst. In any case, an appropriate design for the control strategy should minimize the external fuel requirements at the furnace and enable operation far from the saturation of the final control elements.

*2.2. Control Strategy Results*

2.2.1. Control Strategy One: Single-Loop Feedback Control

In a previous article, the authors explored the closed-loop operation of a reactor/FEHE system for VOC abatement [19]. It was demonstrated that a single-loop feedback control structure (control strategy one) would be able to prevent the limit-cycle scenario by means of the manipulation of a bypass flow for the feed stream around the FEHE. When VOC concentration disturbances occurred, the control system proved to be successful in maintaining the reactor inlet temperature at its set point, thus enabling stable operation and respecting the emission limits (ELV).

The discharge of industrial gases contaminated with VOCs is characterized by variable emission patterns that can fluctuate within wide ranges. In this section, the response of the single-loop feedback control structure to the most frequent disturbance variables—$C_{0Et}$ and $F_{V0}$—is evaluated

The initial operating condition for the base case is shown in Figure 4a,b: $C_{0Et}$ = 600 mg C/Nm$^3$, $F_{V0}$ = 10,000 Nm$^3$/h, and $T_{0,sp}$ = 185 °C. At $t$ = 1000 s, a negative step change occurs in $C_{0Et}$ (from 600 down to 400 mg C/Nm$^3$). After 1 h (3600 s) of operation, a second positive step change occurs in the feed flowrate (from 10,000 to 15,000 Nm$^3$/h).

Figure 4b–d show the response of the system to these disturbances when control strategy one is implemented. Once the $C_{0Et}$ concentration drops from 600 to 400 mg C/Nm$^3$, the heat generation inside the reactor decreases, leading to a lower outlet reactor temperature, $T_e$ (Figure 4b). A higher fraction of the feed stream passes through the heat exchanger; i.e., $\lambda$ increases from 0.66 to 0.712 (Figure 4c) to maintain $T_0$ at its reference value, $T_{0,sp}$ = 185 °C. After this first disturbance, the process still remains within the emission limit ($C_{VOCs}$ < ELV; i.e., $C_{Et} + C_{Ac}$ < 20 mg C/Nm$^3$ at the reactor outlet), as demonstrated in Figure 4d.

This state is maintained until the second disturbance in the feed flowrate occurs. The sudden increase in $F_{V0}$ leads to a further increase in the value of $\lambda$ ($\approx$ 0.87) to maintain the temperature set point. However, the VOC conversion decreases due to the lower residence time of the gas stream inside the reactor and the system cannot satisfy the ELV requirement for the VOCs (Figure 4d).

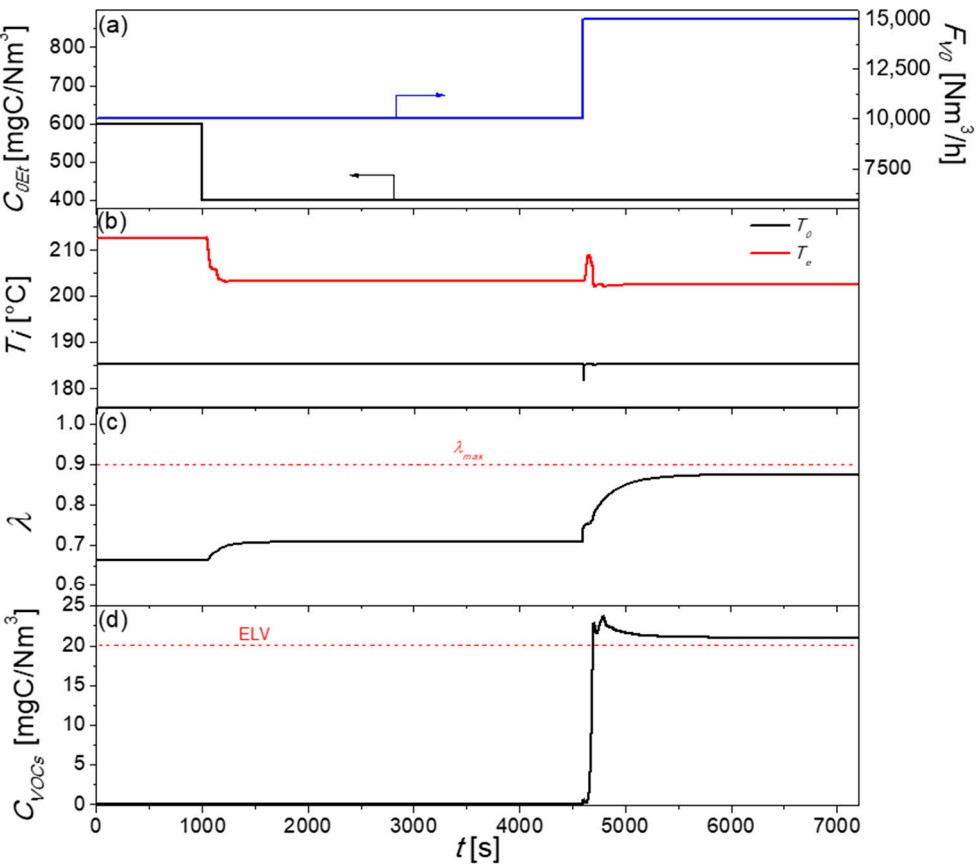

**Figure 4.** (**a**) Successive step disturbances in the inlet ethanol concentration, $C_{0Et}$, and the feed flowrate, $F_{V0}$. (**b**) Temporary profiles for inlet and outlet reactor temperatures. (**c**) Temporary profile for the fraction of the feed flowrate through the FEHE, $\lambda$. (**d**) Temporary profiles for VOC concentration at the reactor outlet ($C_{VOCs} = C_{Et} + C_{Ac}$). Control strategy one.

Although no loss of controllability is observed in this case, the process operates close to the upper limit of the final control element ($\lambda = 0.9$). Further disturbances forcing the closing off of the bypass valve would unavoidably lead to controllability loss.

Notice that the instantaneous step change in $F_{V0}$ causes a very short negative pulse in $T_0$. As a direct consequence, the reactor outlet temperature ($T_e$) shows an inverse response typical of the dynamics of fixed-bed and monolith catalytic reactors.

Figure 5 shows the axial profiles for the VOC concentration corresponding to the initial and final steady states shown in Figure 4. Successive stepwise disturbances in the feed ethanol concentration ($C_{0Et}$) and feed flowrate ($F_{V0}$) lead to an undesirable VOC leakage, which is unavoidable with the proposed control strategy. A more advanced control strategy appears to be necessary in order to prevent scenarios where the system reactor/FEHE/furnace is unable to fulfil its main goal: eliminating the organic pollutants before venting the air stream.

### 2.2.2. Control Strategy Two: Single-Loop Feedback System with Servomechanism Changes in $T_0$

In this control structure, the regulator scheme of the bypass valve (previously shown) is combined with servomechanism changes in $T_{0,sp}$ calculated by means of a mathematical model as a function of $C_{0Et}$ and $F_{V0}$ (see Equation (6)) with the aim of achieving total VOC conversion. The servomechanism changes in $T_0$ take place 10 min after the disturbance occurs.

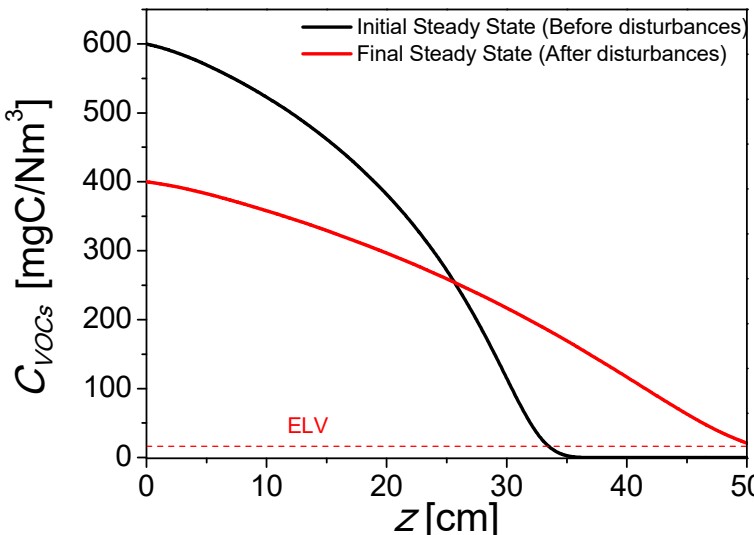

**Figure 5.** Axial profiles for VOC concentration in the reactor corresponding to the initial ($C_{0Et}$ = 600 mg C/Nm$^3$ and $F_{V0}$ = 10,000 Nm$^3$/h) and final ($C_{0Et}$ = 400 mg C/Nm$^3$ and $F_{V0}$ = 15,000 Nm$^3$/h) steady states indicated in Figure 4. Control strategy one.

To test control strategy two, the same disturbances as those applied in control strategy one were analysed (see Figure 6a). The temporary response of the system is shown in Figure 6b–d. As in control strategy one, when the $C_{0Et}$ drops from 600 to 400 mg C/Nm$^3$, heat generation inside the reactor decreases, leading to a lower outlet reactor temperature (Figure 6b) and a higher value for $\lambda$, which increases from 0.66 to 0.712 (Figure 6c) to maintain $T_0$ at its first reference value, $T_{0,sp}$ = 185 °C. Ten minutes after the occurrence of the disturbance, the model calculates a new reference value, $T_{0,sp}$ = 187 °C, and $\lambda$ shows a slight increase to keep $T_0$ at its new set-point value. The positive step change in $T_0$ causes the expected inverse (negative) response in $T_e$ after 200 s (Figure 6b).

When the second disturbance occurs (the step increase in $F_{V0}$), the value of $\lambda$ tends to increase. After 10 min of evolution of $\lambda$, the servomechanism model calculates a new value for $T_{0,sp}$ which leads to a further additional increase in $\lambda$. As can be observed in Figure 6c, $\lambda$ remains at its upper limit value for 500 s. The temporary controllability loss leads to oscillatory behaviour in $T_0$ and $T_e$ (analogous to that shown in Figure 3). Finally, the system stabilizes at a new, higher thermal level ($T_{0,sp}$ = 190 °C). Once this occurs, the controllability of the bypass valve recovers and it sets a value for $\lambda$ slightly lower than the upper limit value. It is clear that further disturbances causing additional closings of the bypass valve would unavoidably lead to controllability loss.

The axial profiles for the VOC concentration presented in Figure 7 show how, with the same successive stepwise perturbations in the inlet ethanol concentration, $C_{0Et}$, and the feed flowrate, $F_{V0}$, control strategy two avoids VOC leakages once the final steady state is reached, demonstrating an improvement with respect to control strategy one. This result can also be confirmed from Figure 6d; that is, $C_{VOC}$ is always below the ELV at the expense of $\lambda$ reaching a final value close to its maximum. To prevent this, it seems necessary for the furnace to supply additional amounts of heat.

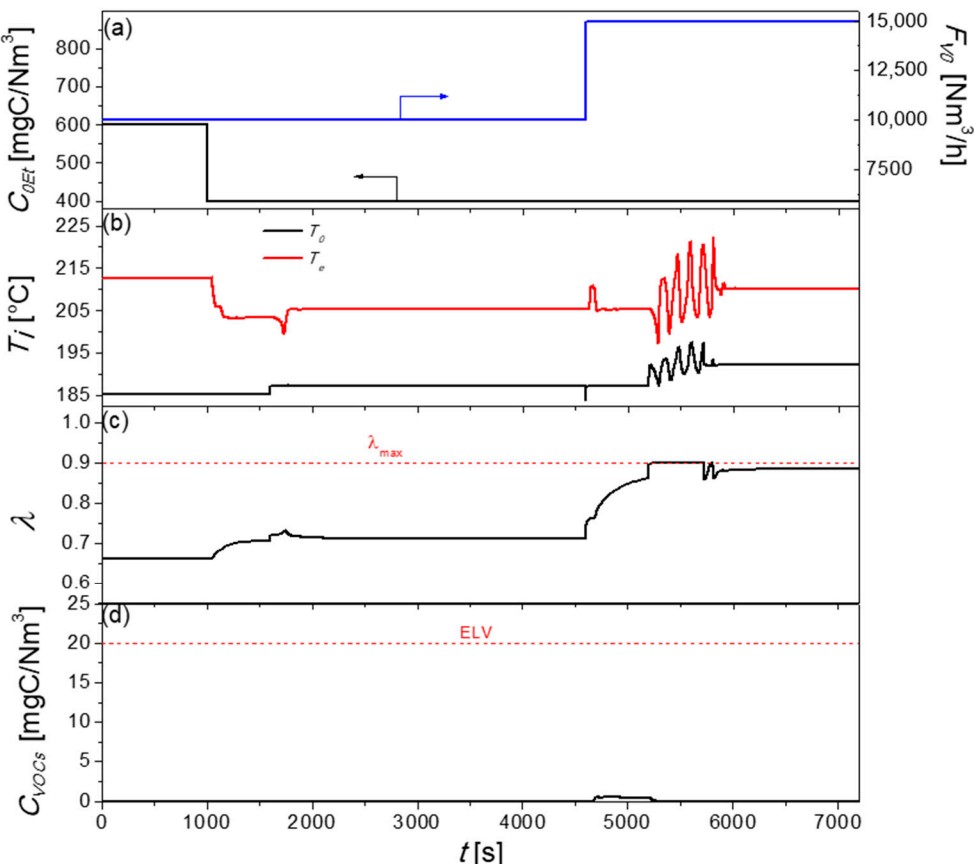

**Figure 6.** (**a**) Successive step disturbances in the inlet ethanol concentration, $C_{0Et}$, and the feed flowrate, $F_{V0}$. (**b**) Temporal profiles for inlet and outlet reactor temperatures. (**c**) Temporal profile for the fraction of the feed flowrate through the FEHE, $\lambda$. (**d**) Temporal profiles for VOC concentration at the outlet ($C_{VOCs} = C_{Et} + C_{Ac}$). Control strategy two.

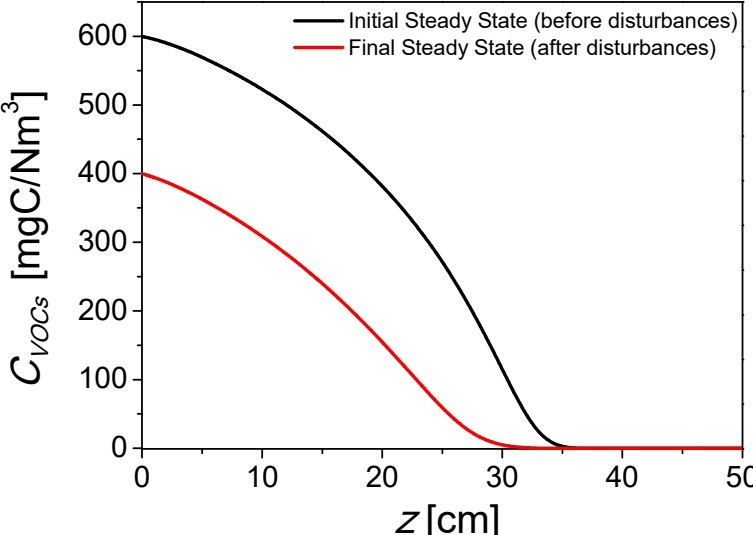

**Figure 7.** Axial profiles for VOC concentration corresponding to the initial steady state ($C_{0Et} = 600$ mg C/Nm$^3$ and $F_{V0} = 10{,}000$ Nm$^3$/h) and final steady state ($C_{0Et} = 400$ mg C/Nm$^3$ and $F_{V0} = 15{,}000$ Nm$^3$/h) indicated in Figure 6a for control strategy two.

### 2.2.3. Control Strategy Three: Sequential Control System

In this control strategy, the previous control structure is combined with a secondary manipulated variable, the natural gas flowrate in the furnace ($F_{NG}$), to reject long-term disturbances and keep the position of the bypass valve (V1) away from saturation positions (closing or full opening) that may lead to controllability loss. The safety margin for $\lambda$ comprises values between 0.6 and 0.7. In cases where the bypass valve (V1) exceeds (over- or underruns) these allowable values for $\lambda$ over 1200 s, a new value for the natural gas flowrate will be set according to a mathematical model (see Equation (7)).

As before, control strategy three was analysed facing the same successive step disturbances (Figure 8a). The temporary responses of the system are shown in Figure 8b–d.

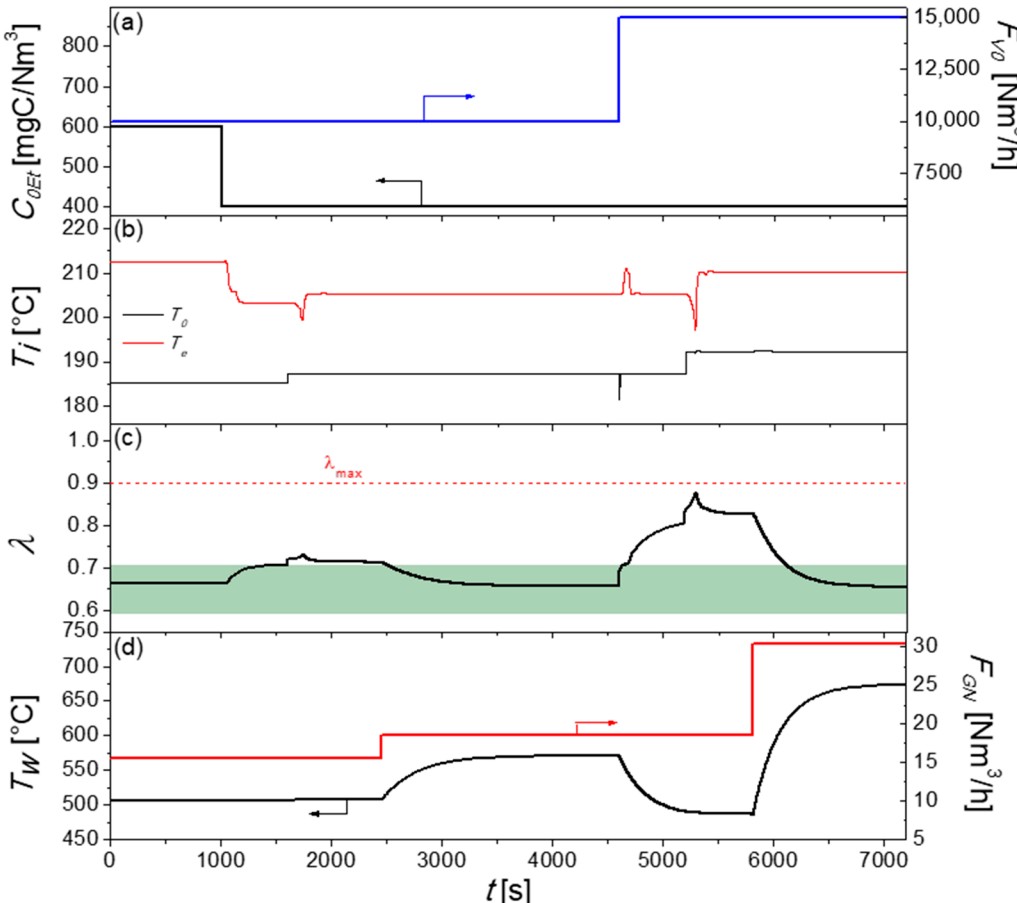

**Figure 8.** (**a**) Successive step disturbances in the inlet ethanol concentration, $C_{0Et}$, and the feed flowrate, $F_{V0}$. (**b**) Temporary profiles for inlet and outlet reactor temperatures. (**c**) Temporary profile for the fraction of the feed flowrate through the FEHE, $\lambda$. (**d**) Temporary profiles for the natural gas flowrate (right ordinate axis) and furnace wall temperature (left ordinate axis).

Due to the first disturbance (negative step change in $C_{0Et}$) and the servomechanism adjustment in $T_0$ (positive step change in $T_{0,sp}$) that takes place after the disturbance, the value of $\lambda$ increases from 0.66 to almost 0.73 (Figure 8c). Once the safety margin for $\lambda$ has been overpassed over a lapse of time of 1200 s, the sequential control strategy leads to an increase in $F_{NG}$ (from 15 to 17 Nm$^3$/h; Figure 8d, right ordinate axis) in order to restore $\lambda$'s initial value of 0.65.

The second disturbance (positive step change in $F_{V0}$) causes a new increase in $\lambda$ from 0.65 to around 0.8. As in the previous case, after 10 min of evolution of $\lambda$, the servomechanism model calculates a new value for $T_{0,sp}$ (190 °C), which leads to a further increase in $\lambda$. After $\lambda$ has been above the upper margin of the band for 20 min, the sequential

control strategy sets a new $F_{NG}$ increase (from 17 to 30 Nm³/h; Figure 8d) to restore the desired value of $\lambda = 0.65$ (Figure 8c).

Notice that, in order to return $\lambda$ to the desired value of 0.65, the furnace wall temperature, $T_w$, increases from an initial value of $T_w = 505\,^\circ\text{C}$ to a final value of 675 °C (Figure 8d). This is a consequence of two facts: first, the need to process a more diluted feed stream with a flowrate 50% higher than that of the base case and, second, the two consecutive increases in the set point to fulfil the process requirements.

Transient profiles for the VOC outlet concentration are not included here. However, as in the case shown in Figure 6d, $C_{VOC}$ is always below the ELV.

The proposed control strategy ensures safe and stable operation, avoiding both temporary emissions of VOCs and controllability loss and keeping the position of the bypass valve (V1) away from saturation positions (closing or full opening).

### 2.3. Comparison of Control Strategies: Energy Savings

Control strategy three also has the potential to minimize the external energy consumption. To demonstrate this, control strategies one and three were compared with step changes in the opposite direction to the previous ones in $C_{0Et}$ and $F_{V0}$. These successive step disturbances lead to an excess of heat in the system and generate the opportunity for energy savings.

The initial steady state is shown in Figure 9a and corresponds to the operative condition for the base case: $C_{0Et} = 600$ mg C/Nm³, $F_{V0} = 10{,}000$ Nm³/h, and $T_{0,sp} = 185\,^\circ\text{C}$. At $t = 1000$ s, a positive step change occurs in $C_{0Et}$ (from 600 to 800 mg C/Nm³). After 60 min (3600 s) of operation, a successive negative step change occurs in the feed flowrate (from 10,000 to 5000 Nm³/h).

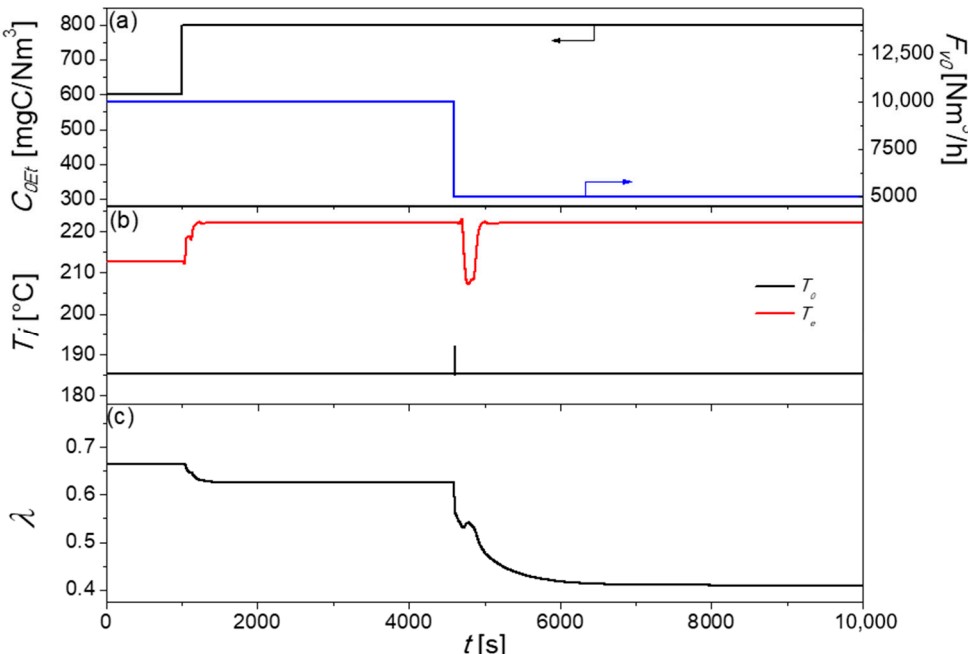

**Figure 9.** (**a**) Successive step disturbances in the inlet ethanol concentration, $C_{0Et}$ (from 600 to 800 mg C/Nm³), and in the feed flowrate, $F_{V0}$ (from 10,000 to 5000 Nm³/h). (**b**) Temporal profiles for the inlet and outlet reactor temperatures. (**c**) Temporal profile for the fraction of the feed flowrate through the FEHE, $\lambda$. Control strategy one.

Figure 9b,c show the response of the system to these disturbances when control strategy one is implemented. Once the $C_{0Et}$ concentration rises from 600 to 800 mg C/Nm³, heat generation inside the reactor increases, leading to an increase in the outlet reactor temperature, $T_e$, from 212 °C to 222 °C (Figure 9b) and a lower fraction of the feed stream being

required through the heat exchanger to maintain $T_0$ at its reference value, $T_{0,sp} = 185\ °C$; $\lambda$ decreases from 0.66 to 0.62 (Figure 9c).

When the second disturbance (negative step change in $F_{V0}$) takes place, a further decrease in the value of $\lambda$ ($\approx$0.4) is required to achieve the set point in $T_0$. Using this simple control strategy, the outlet VOC concentration is maintained below the ELV (results not shown) without any loss of controllability. However, a final value of $\lambda \approx 0.4$ means that around 60% of the feed stream is bypassing the FEHE; that is, the heat exchanger is operating far below its maximum heat-recovery capacity. As a consequence, the air stream is vented to the atmosphere at a higher temperature (an increase in the variable $T_{2e}$, indicated in Figure 1, occurs).

Control strategy three was studied for the same successive step disturbances (Figure 10a). The temporary response of the system is shown in Figure 10b–d.

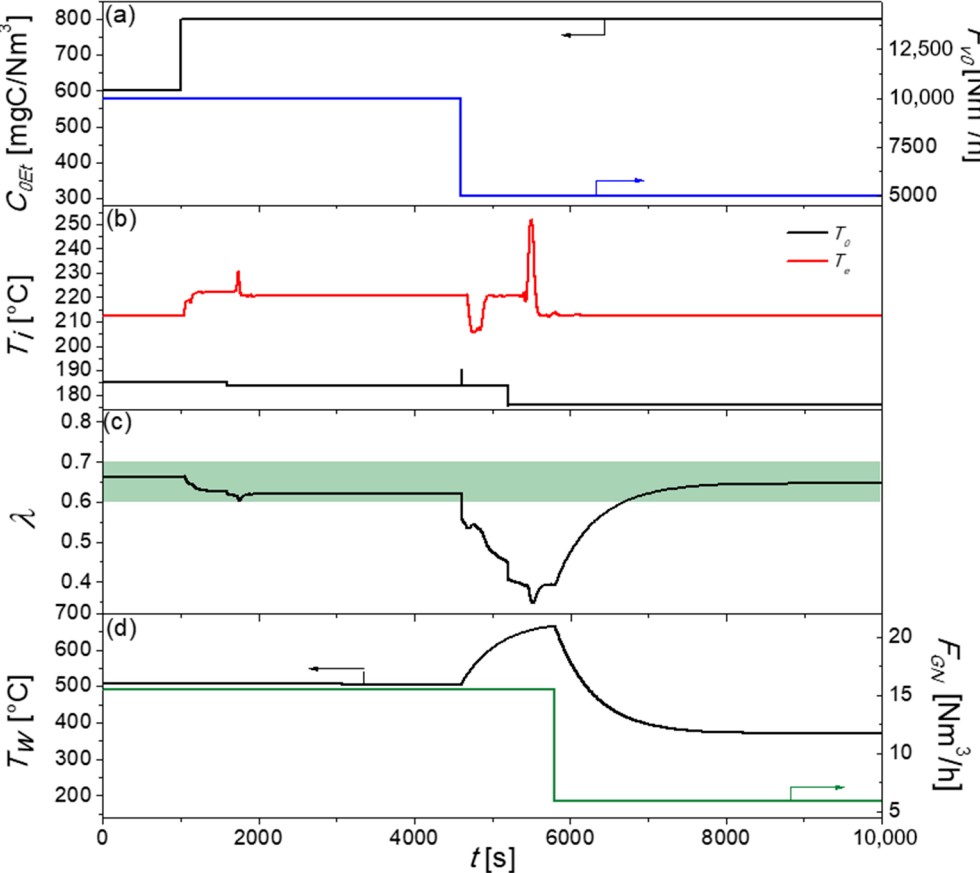

**Figure 10.** (**a**) Successive step disturbances in the inlet ethanol concentration, $C_{0Et}$ (from 600 to 800 mg C/Nm$^3$), and in the feed flowrate, $F_{V0}$ (from 10,000 to 5000 Nm$^3$/h). (**b**) Temporary profiles for the inlet and outlet reactor temperatures. (**c**) Temporary profile for the fraction of the feed flowrate through the FEHE, $\lambda$. (**d**) Control strategy three.

After the first disturbance (positive step change in $C_{0Et}$), the increase in the outlet reactor temperature, $T_e$, causes a decrease in the fraction of the feed flowrate that passes through the FEHE. Within 10 min after the disturbance, the servomechanism adjusts $T_0$ through a slight negative step change in $T_{0,sp}$, which leads to a new decrease in $\lambda$. Despite this, $\lambda$ is maintained within the safety margin imposed, and the sequential control on the natural gas valve does not turn on. The second negative step disturbance in the feed flowrate causes a new servomechanism change in $T_0$ (after 10 min), and as a result of both changes, $\lambda$ decreases twice consecutively and leaves the band of admissible values for 1200 s (20 min). After that, the sequential control strategy sets an $F_{NG}$ decrease (from

its initial value of 15 to 6 Nm$^3$/h; Figure 10d) to restore $\lambda$ to the desired value of 0.65 (Figure 10c), and a significant reduction in the external energy consumption is achieved.

The furnace wall temperature, $T_w$, increases as a result of the decrease in $F_{V0}$ and decreases once $F_{NG}$ has dropped due to the sequential action (Figure 10d). This is another advantage of control strategy three, allowing it to prevent unnecessary overheating of the tubes in the furnace.

Finally, Figure 11 shows the axial VOC concentration profiles for the common initial steady state and the two final steady states achieved with both control strategies. These strategies achieve total VOC conversion but, in control structure three, the servomechanism changes in $T_0$ enable more distributed use of the catalyst along the reactor length. In contrast to CS1 (red curve), in the case of CS3 (green curve), the reaction does not proceed only in a narrow zone close to the entrance of the monolithic reactor.

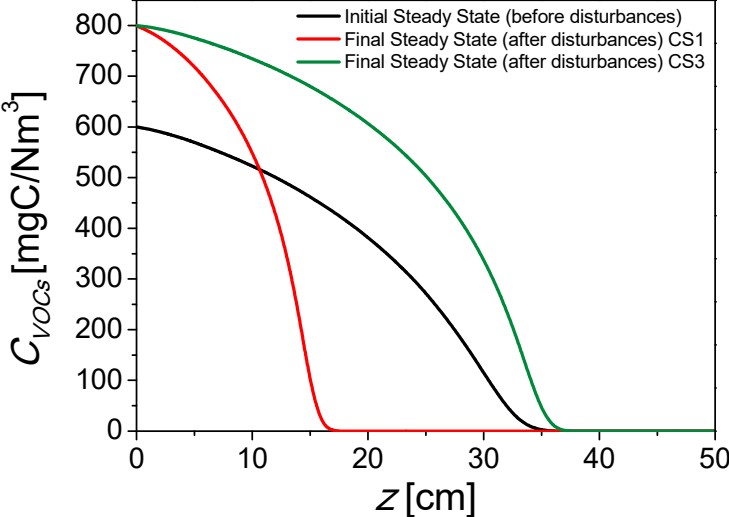

**Figure 11.** Axial profiles of outlet VOC concentration in the initial steady state ($C_{0Et}$ = 600 mg C/Nm$^3$ and $F_{V0}$ = 10,000 Nm$^3$/h) and final steady state ($C_{0Et}$ = 800 mg C/Nm$^3$ and $F_{V0}$ = 5000 Nm$^3$/h) for control strategies one and three.

## 3. Mathematical Model

### 3.1. Description of the Process and Control Strategies

For the closed-loop analysis, three different control system strategies were analysed with a dual purpose: ensuring safe and stable operation and avoiding temporary emissions of VOCs.

#### 3.1.1. Single-Loop Feedback Control

Figure 12 shows the schematic representation of the process under study. A monolithic reactor is coupled with an upstream feed-effluent heat exchanger and a natural gas furnace. The energy of the hot reactor effluent is recycled back to the reactor through the FEHE, where a fraction $\lambda$ of the feed stream (the VOC-contaminated gas stream at room conditions) is preheated. The other fraction of the feed $(1 - \lambda)$ is bypassed around the FEHE and mixed with the stream leaving the FEHE. Downstream of the mixer, the feed stream is sent to the furnace, where it receives the additional heat needed to reach the reactor inlet temperature, $T_0$. In this case, a single-loop feedback control system is used to maintain the temperature at the set point ($T_{0,sp}$) by handling the bypass valve. Since the reactor inlet temperature is controlled by manipulating the bypass flow, a constant value for the heat duty in the furnace is assumed. Two split-range valves (V1 and V2) are used to manipulate the bypass flowrate and the flowrate through the FEHE with the aim of achieving the required fluid mechanics [20]. In this closed-loop structure, $T_{0,sp}$ remains constant.

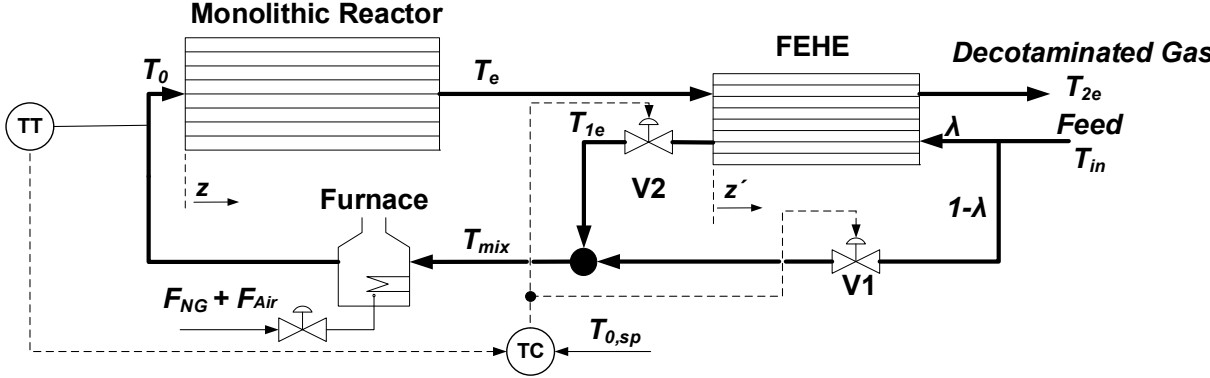

**Figure 12.** Schematic representation of a single-loop feedback control system for the reactor/FEHE/furnace process.

3.1.2. Single-Loop Feedback System with Servomechanism Changes in $T_0$.

Figure 13 schematizes a single-loop feedback system that allows servomechanism changes in $T_{0,sp}$. The $T_{0,sp}$ values set by the model are obtained through steady-state simulations of the monolithic reactor in order to achieve total VOC conversion in 70% of the reactor length with wide operating ranges for two measurable variables: $F_{V0}$ (an inlet variable) and $\Delta T$ in the reactor (an output variable). $\Delta T = T_e - T_0$ becomes $\Delta T_{ad}$ under conditions of total conversion of VOCs. Since $\Delta T_{ad}$ is proportional to the inlet concentration of ethanol ($C_{0Et}$), $\Delta T$ can be considered as an indirect measurement of $C_{0Et}$. Although the changes in $\Delta T$ are delayed with respect to the corresponding changes in $C_{0Et}$, this approach avoids the need to set a continuous measurement of the VOC concentration in the feed stream.

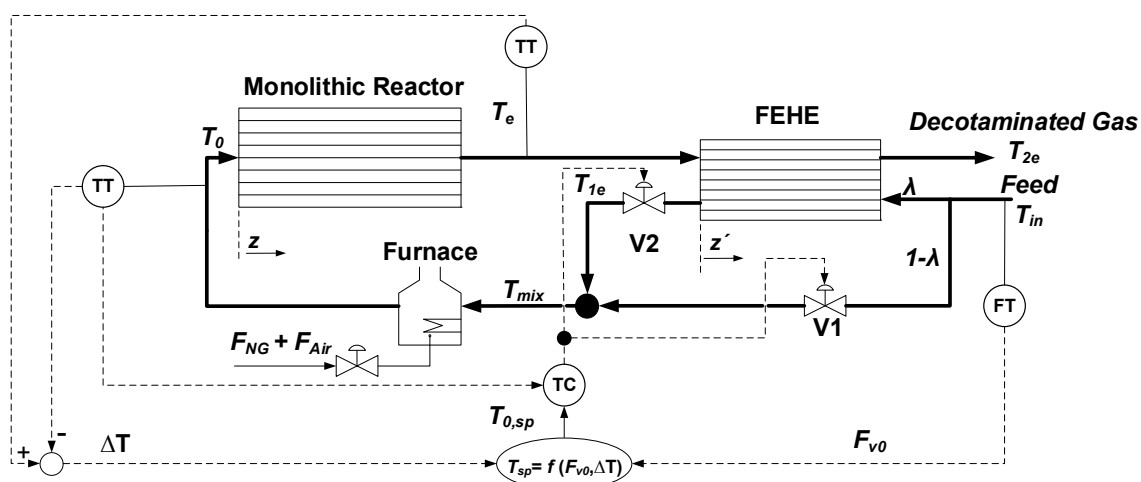

**Figure 13.** Schematic representation of a single-loop feedback control system with servomechanism changes in $T_0$.

The $T_{0,sp}$ stated by the model guarantees total VOC conversion inside the reactor. The proposed control system enables servomechanism changes in $T_0$ (output variable) with the aim of avoiding situations involving temporary emissions of VOCs due to high feed flowrates (short residence times in the reactor) and/or low inlet VOC concentrations for which the value of $T_{0,sp}$ would need to be increased [18,21]. In this control structure, the regulatory scheme of the bypass valve (previously shown) maintains the $T_0$ required to face any other disturbance.

### 3.1.3. Sequential Control System

Figure 14 shows the structure of an advanced control strategy (a sequential control system) that consists of a single-loop feedback system with servomechanism changes in $T_0$ to reject short-term disturbances followed by a secondary manipulated variable, the natural gas flowrate in the furnace ($F_{NG}$), to reject long-term disturbances. The objective of this secondary manipulated variable is to keep the position of the bypass valve (V1) within a convenient band, away from saturation positions (closing or full opening) that may lead to controllability loss. This valve position is represented here by the variable $\lambda$; i.e., the fraction of the feed flow that passes through the FEHE.

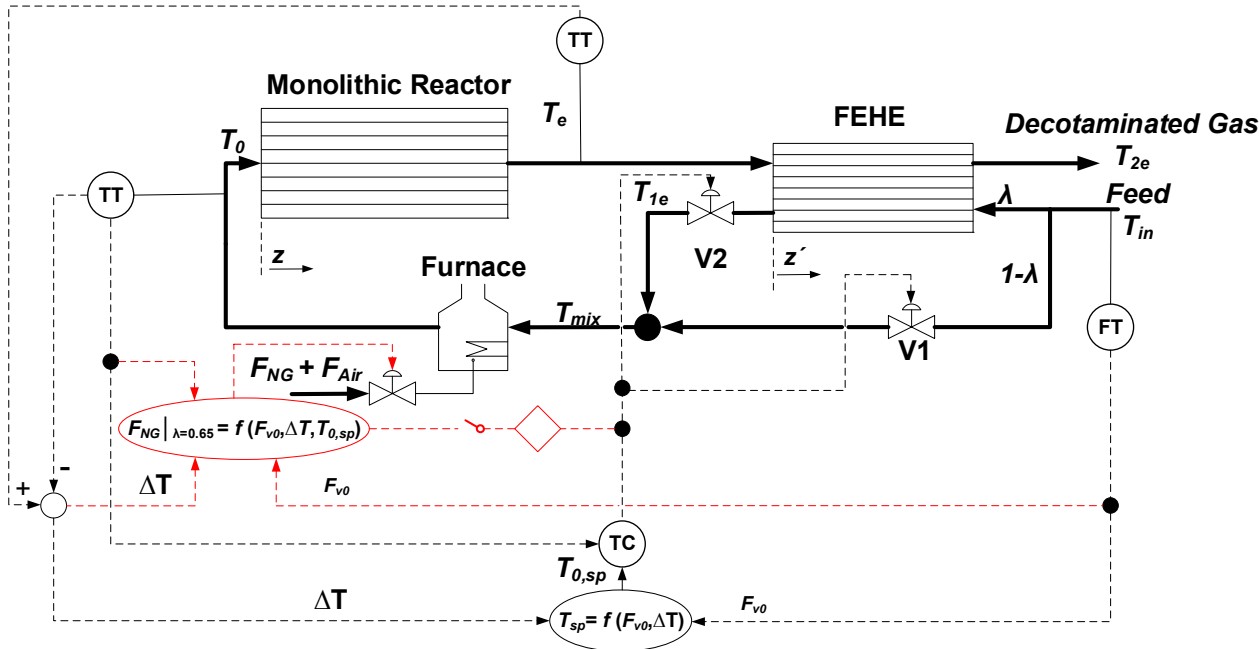

**Figure 14.** Schematic representation of a sequential control system (advanced control system).

The model used to calculate the natural gas flowrate in the furnace ($F_{NG}$) is obtained by using steady-state simulations from measurable variables, such as the feed flowrate ($F_{V0}$), the adiabatic temperature rise in the reactor ($\Delta T = T_e - T_0$), and the set-point inlet temperature ($T_{0,sp}$), making it possible to determine the desired value for the opening of the valve V1 that would lead to a value of $\lambda \approx 0.65$.

In cases where the bypass valve (V1) exceeds (over- or underruns) a convenient band of admissible values for $\lambda$ over a sufficiently long period of time (1200 s), the red connection in Figure 14 will switch on, activating the feedforward loop.

### 3.2. Reactor, FEHE, and Furnace

To simulate the non-steady-state operation of the adiabatic monolithic reactor, a 1D pseudohomogeneous plug-flow model was developed. Table 1 shows the reaction system considered, which included the partial oxidation of ethanol to acetaldehyde (reaction one) and the total oxidation of acetaldehyde (reaction two). The kinetic expressions developed by Campesi et al. (2011) [22] for the Mn-Cu catalyst were adopted. Table 2 shows the kinetic parameters that were adjusted to obtain observable expressions for the reaction rate, considering internal and external resistances to mass transfer and external resistances to heat transfer [23].

**Table 1.** Reaction system and kinetic expressions [22].

| Reaction System | Kinetic Expression | |
|---|---|---|
| $C_2H_6O + (1/2)\,O_2 \rightarrow C_2H_4O + H_2O$ | $r_1 = \dfrac{k_{ref1}\exp\left[-(E_1/R)\left(1/T - 1/T_{ref}\right)\right]C_{Et}}{1 + Kc_{Et}C_{Et} + Kc_{Ac}C_{Ac}}$ | (1) |
| $C_2H_4O + (5/2)O_2 \rightarrow 2CO_2 + 2H_2O$ | $r_2 = \dfrac{k_{ref2}\exp\left[-(E_2/R)\left(1/T - 1/T_{ref}\right)\right]Kc_{Ac}C_{Ac}}{1 + Kc_{Et}C_{Et} + Kc_{Ac}C_{Ac}}$ | (2) |

**Table 2.** Effective kinetic parameters and standard heats of reaction [23].

| Parameter | Value |
|---|---|
| $k_{ref,1}$ | $1.71 \times 10^3$ s$^{-1}$ |
| $k_{ref,2}$ | $1.8 \times 10^{-1}$ mol s$^{-1}$ m$^{-3}$ |
| $E_1$ | $1.1124 \times 10^5$ J mol$^{-1}$ |
| $E_2$ | $1.62 \times 10^5$ J mol$^{-1}$ |
| $Kc_{Ac}$ | $6.75 \times 10^2$ m$^3$ mol$^{-1}$ |
| $Kc_{Et}$ | ~0 |
| $\Delta H_{0r1}$ | $-1.73 \times 10^5$ J/mol |
| $\Delta H_{0r2}$ | $-1.10 \times 10^6$ J/mol |

Tables 3 and 4 show the operating conditions and geometric parameters for the system. The mass balances in the reactor, as well as the energy balances in the reactor and FEHE, are expressed through partial differential equations. The axial coordinate is discretized in both devices by means of backward first-order finite differences. The resulting set of ordinary differential equations, together with the global energy balances in the furnace (for the gas phase and tube wall), are solved simultaneously by integration over time using a Gear algorithm [24]. The details of the models can be found in Miranda et al. (2023) [18].

**Table 3.** Process operating conditions.

| Parameter | Value |
|---|---|
| Feed VOC concentration, $C_{Et0}$ | 400–800 mgC/Nm$^3$ |
| Volumetric feed flowrate, $F_{V0}$ | 5000–15,000 Nm$^3$/h |
| Pressure, $P$ | 101.325 kPa |
| Feed temperature, $T_{In}$ | 20 °C |
| Reactor gas hourly space velocity, $GHSV$ | $5.93 \times 10^3$–$1.798 \times 10^4$ 1/h |
| Flow fraction through FEHE, $\lambda$ | 0.1–0.9 |
| Furnace heat, $Q_f$ | 50–300 kW |

### 3.3. Controller

A proportional–integral (PI) controller was selected for the manipulation of the bypass valve in the three control strategies under consideration:

$$(1 - \lambda) = \overline{(1 - \lambda)} + K_c\left(e_r + \frac{1}{\tau_i}\int_0^t e_r dt\right) \tag{3}$$

The tuned parameters of the PI controller are listed in Table 5 [19]. Strong integral action is required for efficient control over this kind of process with positive heat feedback [16].

**Table 4.** Geometrical parameters.

| Parameter | Value |
|---|---|
| Reactor | |
| Channel length, $L$ | 0.5 m |
| Channel width = height, $b$ | 1115 um |
| Cell density | 400 cpsi |
| Channel number | $13{,}924 \times 102$ |
| Monolithic material | Cordierite ($2MgO\text{-}2Al_2O_3\text{-}5SiO_2$) |
| Catalytic material | Mn-Cu |
| Average washcoat thickness | 20 μm |
| Washcoat density, $\rho_w$ | 4030 kg/m$^3$ |
| Reactor weight (washcoat + cordierite) | 682 kg |
| Heat exchanger [25] | |
| Type | Plate fin heat exchanger (PFHE) |
| Flow configuration | Counter-current |
| Plate length, $L'$ | 0.7 m |
| Plate width | 0.7 m |
| Channel height | 6.35 mm |
| Plate thickness | 0.4 mm |
| Plates number | 220 |
| Fin thickness | 0.154 mm |
| Fin density | 437 fins/m |
| FEHE weight | 903 kg |
| Heat exchanger material | Stainless steel (AISI 316) |
| Furnace | |
| Type | Indirect fired |
| Tube arrangement | Horizontal |
| Tube number | 98 + chamber |
| Tube length | 1000 mm |
| Tube/chamber diameters | 25.4 mm/400 mm |
| Fuel | Natural gas |
| Furnace weight | 260 kg |
| Efficiency, $\varepsilon_f$ | 0.9 |

**Table 5.** Parameters of the PI controller [19].

| Controller Parameter | Value |
|---|---|
| Proportional action parameter, $K_c$ | 0.003 |
| Integral action parameter, $\tau_i$ | 0.5 |

### 3.3.1. Control Strategy One: Single-Loop Feedback System

The error for the controlled variable used in Equation (3) is calculated as follows:

$$e_r = T_0 - T_{0,sp} \tag{4}$$

where $T_{0,sp}$ is a constant value.

### 3.3.2. Control Strategy Two: Single-Loop Feedback System with Servomechanism Changes in $T_0$

In this case, the value of $T_{0,sp}$ is predicted from measurable variables—the feed flowrate ($F_{V0}$) and the temperature rise in the reactor ($\Delta T = T_e - T_0$)—in order to maintain the total VOC consumption across 70% of the reactor length.

$$e_r = T_0 - T_{0,sp} \tag{5}$$

$$T_{0,sp} = f(F_{V0}, \Delta T) \text{ to satisfy that: } X_{VOCs} = 1 \text{ at } z \approx 0.70L \tag{6}$$

### 3.3.3. Control Strategy Three: Advanced Control System

As mentioned above, the proposed advanced control system employs sequential control. It consists of a single-loop feedback system with a main manipulated variable followed by a secondary manipulated variable, the natural gas flowrate in the furnace ($F_{NG}$). $F_{NG}$ is predicted from measurable variables—the feed flowrate ($F_{V0}$), the temperature rise in the reactor ($\Delta T = T_e - T_0$), and the set-point inlet temperature ($T_{0,sp}$)—to obtain a desired value for the opening of the valve V1 that leads to a value of $\lambda \approx 0.65$.

$$F_{NG}\big|_{\lambda \approx 0.65} = f(\Delta T, F_{V0}, T_{0,sp}) \tag{7}$$

In Equation (7), $T_{0,sp}$ is not a fixed value but is calculated using Equation (6).

### 4. Conclusions

A novel control structure was proposed consisting of a single-loop feedback system with servomechanism changes in the inlet reactor temperature that manipulates the bypass valve around the FEHE and, sequentially, the natural gas flowrate in the furnace whenever the bypass value is outside the imposed operating limits for extended periods of time.

The control strategy makes it possible to satisfy the main control objective (VOC abatement) by avoiding situations involving loss of controllability due to saturation (closure) of the bypass valve when almost all the feed has to be redirected to the FEHE. In addition, it minimizes the consumption of natural gas when the feed stream is more concentrated in VOCs or the feed flowrate decreases. This prevents excessive overheating of the tube wall in the furnace.

**Author Contributions:** Conceptualization, M.L.R., F.M.S. and D.O.B.; methodology, A.F.M.; software, A.F.M.; validation, A.F.M.; formal analysis, M.L.R. and D.O.B.; investigation, A.F.M.; resources, M.L.R.; writing—original draft preparation, M.L.R. and A.F.M.; writing—review and editing, A.F.M., M.L.R. and D.O.B.; visualization, A.F.M.; supervision, M.L.R. and D.O.B.; project administration, D.O.B.; funding acquisition, M.L.R. and D.O.B. All authors have read and agreed to the published version of the manuscript.

**Funding:** This research was funded by the National University of San Luis, (Argentina, PROICO 14-4318), the National Agency for Scientific and Technical Promotion (ANPCyT) (Argentina, PICT 2017-4516), and the National Scientific and Technical Research Council (CONICET) (Argentina, PIP 1122020010315CO).

**Data Availability Statement:** Not applicable.

**Acknowledgments:** The authors acknowledge the support of the National University of San Luis, the National University of the South, the National Agency for Scientific and Technical Promotion, and the National Scientific and Technical Research Council (CONICET).

**Conflicts of Interest:** The authors declare no conflict of interest.

### Nomenclature

| | |
|---|---|
| $b$ | Channel width=height, mm |
| $C_j$ | Concentration of component $j$, $\text{mol}_j/\text{m}^3$ or mg C/Nm$^3$ |
| $e_r$ | Error, dimensionless |
| $E_i$ | Activation energy of reaction $i$, J/mol |
| $ELV$ | Emission limit value, 20 mg C/Nm$^3$ (total VOC emissions under normal conditions) |

| | |
|---|---|
| $F_{NG}$ | Natural gas flow under normal conditions, $Nm^3/h$ |
| $F_{V0}$ | Volumetric feed flowrate, $Nm^3/h$ |
| $F_{Air}$ | Volumetric air flowrate, $Nm^3/h$ |
| $GHSV$ | Gas hourly space velocity, $1/h$ |
| $k_{ref,1}$ | Kinetic constant for reaction one, $1/s$ |
| $k_{ref,2}$ | Kinetic constant for reaction two, $mol/(m^3\ s)$ |
| $Kc_j$ | Adsorption constant for component $j$, $m^3/mol$ |
| $K_c$ | Proportional action parameter |
| $L$ | Reactor channel length, m |
| $L'$ | Heat exchanger length, m |
| $P$ | Pressure, kPa |
| $Q_f$ | Furnace heat supply, kW |
| $r_i$ | Reaction rate for reaction $i$, $mol/(m_g^3 s)$ |
| $t$ | Time, s |
| $T$ | Reactor temperature, $°C$ |
| $T_e$ | Reactor outlet temperature = hot stream inlet temperature (heat exchanger), $°C$ |
| $T_k$ | Temperature of stream $k$ in the heat exchanger, $°C$ |
| $T_w$ | Temperature of furnace tube wall |
| V1 | Split-range valve to manipulate the bypass flowrate |
| V2 | Split-range valve to manipulate the flowrate through the heat exchanger |
| $X_j$ | Conversion of component $j$, dimensionless |
| $X_{VOCs}$ | VOC conversion |
| $Z$ | Reactor axial coordinate, m |
| $z'$ | Heat exchanger axial coordinate, m |

Compounds/acronyms
| | |
|---|---|
| VOC | Volatile organic compound |

Greek letters
| | |
|---|---|
| $\Delta H_{ri°}$ | Heat of reaction $i$ under standard conditions, J/mol |
| $\Delta T$ | Temperature difference between the reactor outlet and the reactor inlet |
| $\Delta T_{ad}$ | Adiabatic temperature gradient, $°C$ |
| $\Delta T_f$ | Temperature rise caused by the furnace, $°C$ |
| $\Delta C_{0Et}$ | Concentration disturbance, mg C/$Nm^3$ |
| $\varepsilon_1$ | Degree of advancement of reaction one |
| $\varepsilon_2$ | Degree of advancement of reaction two |
| $\varepsilon_f$ | Furnace thermal efficiency |
| $\lambda$ | Fraction of stream through the heat exchanger |
| $1-\lambda$ | Fraction of stream through bypass |
| $\tau_i$ | Integral action parameter |

Subscripts
| | |
|---|---|
| $Ac$ | Acetaldehyde |
| $e$ | Exit |
| $Et$ | Ethanol |
| $f$ | Furnace |
| $w$ | Furnace tube wall |
| $HE$ | Heat exchanger |
| $i$ | Reaction |
| $j$ | Component |
| $mix$ | Mixing point |
| $R$ | Reactor |
| $sp$ | Set point (reference value) |
| $VOCs$ | Volatile Organic Compounds (ethanol + acetaldehyde) |
| $T$ | Total |
| 1 | Heat exchanger cold stream |
| 2 | Heat exchanger hot stream |

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
