# Peer review of "Novel Control System Strategy for the Catalytic Oxidation of VOCs with Heat Recovery"

_catalysts, doi:10.3390/catal13050897_

Round 1

Reviewer 1 Report

This is an excellent and very complete work on the development of control systems for structured catalytic reactor systems (monoliths) coupled to heat exchangers for the removal of volatile pollutants.

The mathematical rigour in the approach and resolution of the control model is excellent and the cases studied are relevant. For all these reasons, the work deserves to be published in its current format.

Author Response

Dear reviewer,

Thank you for your positive comments about our work particularly on the mathematical rigor of our approach and the relevance of the case studies presented.

We are pleased to inform you that we have submitted a revised version of the manuscript. We hope the new version will further improve the clarity and quality of our work.

Best regards.

Reviewer 2 Report

The manuscript reports the analysis of several control strategies for VOC oxidation reactors with heat integration. The topic is of high relevance in view of the very steep changes in inlet conditions that these systems can experience, being operated in industrial environments, therefore proper control systems are required. The system presents several strategies and clearly illustrates the advatages of the developed methodology.The paper is well written and results are scientifically sound. I therefore recommend the paper for publication in the present form.

Author Response

Dear reviewer,

Thank you for your positive feedback on our manuscript. We are pleased to hear that you find our work relevant and well-written, and that the results are scientifically sound.

We are pleased to inform you that we have submitted a revised version of the manuscript. We hope the new version will further improve the clarity and quality of our work.

Best regards.

Reviewer 3 Report

In the submitted manuscript, configurations of a system consisting of reactor, feed-effluent heat exchanger and furnace for operation in the catalytic combustion of volatile organic compounds are theoretically discussed. Efforts were made to propose a solution that would ensure a safe and stable operation. Of the three presented models, the single-loop feedback system with servomechanism changes at the inlet reactor temperature presented the best results. When I red this paper my first impression was strongly negative towards the journal to which it is directed. The authors did not put any emphasis on the catalytic aspects of the process. They did not even obtain their own experimental data for the work of Mn-Cu catalyst in the combustion of acetaldehyde and ethanol. The kinetic parameters were taken from ref. [20]. Nevertheless, after thinking about it, I came to the conclusion that the presented results may be of some interest to researchers specializing in catalysis. Therefore, I suggest publishing this work after thorough linguistic proofreading. Both the grammar and the numerous typos in the text and even in the figures require correction.

The manuscript requires a thorough linguistic correction.

Author Response

Dear reviewer,

Thank you for your critical revision of our article.

We apologise for the grammatical and typographical errors in the original manuscript and appreciate your recommendation for a thorough linguistic revision. The revised manuscript has undergone a thorough editing process to improve the clarity and quality of our work.

We value your input and appreciate the opportunity to improve our manuscript.

Kind regards.